A biomechanical analysis of the stand-up paddle board stroke: a comparative study

http://orcid.org/0000-0002-1865-0488 Schram Ben 1 2
http://orcid.org/0000-0001-7773-0253 Furness James 1 2
Kemp-Smith Kevin 1 2 kkempsmi@bond.edu.au
Sharp Jason 2
Cristini Matthew 2
http://orcid.org/0000-0001-7693-4158 Harvie Daniel 2
Keady Emma 2
Ghobrial Maichel 2
Tussler Joshoa 2
Hing Wayne 1 2
Nessler Jeff 3
Becker Matthew 3
1 Water Based Research Unit, Bond Institute of Health and Sport, Bond University , Robina, QLD , Australia
2 Faculty of Health Science and Medicine, Bond Institute of Health and Sport, Bond University , Robina, QLD , Australia
3 Department of Kinesiology, California State University, San Marcos , San Marcos, CA , USA
Carpes Felipe
Electronic publication date: 2019 Nov 1
Publication date: 2019
Volume: 7
Electronic Location ID: e8006
Received 2019 May 31; Accepted 2019 Oct 7
Copyright: © 2019 Schram et al.
Copyright year: 2019
Copyright holder: Schram et al.
License: This is an open access article distributed under the terms of the Creative Commons Attribution License, which permits unrestricted use, distribution, reproduction and adaptation in any medium and for any purpose provided that it is properly attributed. For attribution, the original author(s), title, publication source (PeerJ) and either DOI or URL of the article must be cited.
License URL: https://creativecommons.org/licenses/by/4.0/

Keywords: Biomechancis, Stand-up paddle, Technique, Motion analysis, Skill level, Aquatic sport, Kinematics, Surfing, Sports injury, Injury

Funding: The authors received no funding for this work.

==============================
Background

Stand-up paddle boarding (SUP) is a rapidly growing global aquatic sport, with increasing popularity among participants within recreation, competition and rehabilitation. To date, few scientific studies have focused on SUP. Further, there is no research examining the biomechanics of the SUP paddle stroke. The purpose of this study was to investigate whether variations in kinematics existed among experienced and inexperienced SUP participants using three-dimensional motion analysis. This data could be of significance to participants, researchers, coaches and health practitioners to improve performance and inform injury minimization strategies.

Methods

A cross-sectional observational design study was performed with seven experienced and 19 inexperienced paddlers whereby whole-body kinematic data were acquired using a six-camera Vicon motion capture system. Participants paddled on a SUP ergometer while three-dimensional range of motion (ROM) and peak joint angles were calculated for the shoulders, elbows, hips and trunk. Mann–Whitney U tests were conducted on the non-normally distributed data to evaluate differences between level of expertise.

Results

Significant differences in joint kinematics were found between experienced and inexperienced participants, with inexperienced participants using greater overall shoulder ROM (78.9° ± 24.9° vs 56.6° ± 17.3°, p = 0.010) and less hip ROM than the experienced participants (50.0° ± 18.5° vs 66.4° ± 11.8°, p = 0.035). Experienced participants demonstrated increased shoulder motion at the end of the paddle stoke compared to the inexperienced participants (74.9° ± 16.3° vs 35.2° ± 28.5°, p = 0.001 minimum shoulder flexion) and more extension at the elbow (6.0° ± 9.2° minimum elbow flexion vs 24.8° ± 13.5°, p = 0.000) than the inexperienced participants.

Discussion

The results of this study indicate several significant kinematic differences between the experienced and inexperienced SUP participants. These variations in technique were noted in the shoulder, elbow and hip and are evident in other aquatic paddling sports where injury rates are higher in these joints. These finding may be valuable for coaches, therapists and participants needing to maximize performance and minimize injury risk during participation in SUP.

Introduction

Stand-up paddle boarding (SUP) is a mixture of surfing and paddle-based sports where the rider balances on a board (~3–5 m long, ~1 m wide) and grips a single-bladed paddle (~2 m long) to propel themselves through the water (Schram, 2015). Previous research has defined the main components of the SUP stroke as; entry, drive and exit of the paddle from the water. The entry phase denotes entry of the paddle into the water, the drive phase is the forceful pulling stroke through the water and exit phase describes the paddle release and withdrawal from the water (Schram, Hing & Climstein, 2015). To date there has been no scientific research analyzing the biomechanics of the paddle stroke in SUP.

Stand-up paddle boarding is an aquatic recreational and sporting activity that is readily accessible to most people, requires minimal equipment, is easy to learn and provides a low impact physical challenge (Schram, 2015). Despite being a relatively new water-based sport, SUP’s popularity has increased globally due to its purported health and fitness benefits such as improvements in fitness, endurance and self-rated quality of life (Schram, Hing & Climstein, 2016b). According to the “2015 Paddlesports Report,” SUP participation has steadily increased in the United States from 1.1 million in 2010 to 2.8 million in 2014 (Outdoor Foundation and The Coleman Company, 2015). SUP is an activity that is suitable for all ages and skill levels, can be practiced on any body of water and is reported to be an ideal activity for a full-body workout (Mei-Dan & Carmont, 2013; Schram, Hing & Climstein, 2017).

Biomechanical analysis in sport allows for modifications to technique in order to maximize power output and minimize injury (Ho, Smith & O’Meara, 2009; Bini & Carpes, 2014). Epidemiological studies of injuries in SUP have revealed that the shoulder/upper arm (32.9%) lower back (14.3%) and elbow/forearm (11.8%) were the most common locations of injuries reported in a study of both competitive and recreational SUP riders (Furness et al., 2017). The importance of technique is highlighted by the fact that less than optimal stroke biomechanics has been associated with both shoulder, elbow and back injuries in the similar sports of kayaking, and outrigger paddling (Hagemann, Rijke & Mars, 2004). In line with epidemiological studies in SUP, the shoulder is also the most commonly injured site in kayaking accounting for in excess of 30% of all paddling injuries (Fiore & Houston, 2001; Abraham & Stepkovitch, 2012; Bell, Carman & Tumilty, 2013). A biomechanical understanding of the SUP stroke may provide direction towards injury minimization within this sport.

Currently, only non-scientific information exists regarding optimal paddling technique for SUP in the form of online media and instructional videos (Dionne, 2014; Stehlik, 2011; Cain, 2015a), and written guides (Cain, 2015b; Wordpress, 2015). Research into stroke biomechanics has been performed in similar aquatic sports including kayaking, canoeing, and dragon boat racing. However, these studies focused on comparisons between skill levels (Kendal & Sanders, 1992; Ho, Smith & O’Meara, 2009; Limonta et al., 2010), genders (Gomes et al., 2015), equipment (Fleming et al., 2012), training paces (Zahalka et al., 2011; Gomes et al., 2015) and dominant vs non-dominant sides (Limonta et al., 2010; Wassinger et al., 2011). The purpose of these previous investigations were to determine mechanisms in which to maximize performance and minimize injury risk. Despite sharing similarities to other aquatic paddling sports, the SUP stroke does have considerable biomechanical differences. Primarily, the participant is standing up and balancing on a board compared to all other paddle sports where the participant is sitting. Therefore, the purpose of this study was to compare the differences in SUP paddle stroke kinematics between experienced and inexperienced participants. The hypothesis is that there will be significant differences between experience levels of paddlers in regard to paddling technique. Findings may assist in identifying optimal stroke mechanics in order to minimize injury occurrence and improve overall performance.

Materials and Methods

Participants

Experienced and inexperienced SUP participants were recruited for the study. Exclusion criteria included a history of current musculoskeletal injuries or cardiovascular disorders that impacted their ability to undertake the trials. Additionally, any participant that had an allergy to adhesive tape was also excluded. Recruitment was conducted through flyers, emails and face-to-face requests with information to participate in a within-participant laboratory biomechanical analysis of the SUP paddle stroke. An explanatory statement was provided to potential participants and a consent form was provided to those interested in being involved in the study. To be classified as experienced, participants were to have had a history of competition at an international, national or state level within the previous 2 years. Participants who engaged in SUP recreationally and had no history of competition of formal training were classified as inexperienced.

In total, 26 SUP participants were recruited (experienced n = 7, four male, three female, 33 ± 7.8 years, 173.9 ± 50.5 cm, 76.5 ± 12.2 kg; inexperienced n = 19, 11 male, eight female, 24.5 ± 2.4 years, 174.1 ± 63.3 cm, 72.9 ± 11.3 kg) for this study. The experienced paddlers, on average trained five times per week on water and twice on land. Participants were invited to attend a single data collection session at the Bond Institute of Health Motion Analysis Laboratory. Permission to conduct the study was granted by the University Human Research Ethics Committee (0000015422) and all participants provided written informed consent prior to participation.

General protocol

A six-camera, passive, three-dimensional motion analysis system at 100 Hz (Vicon; Oxford Metrics, Inc., Yarnton, Oxfordshire, UK) was utilized to track one cm spherical retroflective markers placed over key bony landmarks according to Vicon’s full body plug-in Gait model. This model uses 39 reflective markers (14 mm) diameter to create nine segments. Cameras were strategically placed around the test area to maximize data capture. Prior to data acquisition, the motion capture system was calibrated in accordance with manufacturer recommendations (Vicon; Oxford Metrics, Inc., Yarnton, Oxfordshire, UK) whereby an L-frame calibration wand was used to align the origin of the capture volume with a point on the surface of a specialized SUP ergometer (KayakPro SUPErgo, Miami, FL, USA). The KayakPro SUP ergometer has previously been validated for clinical testing (Schram, Hing & Climstein, 2015). A static trial was undertaken for anatomical landmark calibration for each of the participants (Besier et al., 2003).

Participants undertook a familiarization period prior to testing which involved a 2-min warm up where they self-selected stroke frequency, stance, rate and paddle change over to the opposite side. At completion of the familization period, the participant performed two consecutive paddling trials (left and right side), in a randomized fashion predetermined by a spreadsheet formula (Microsoft Excel v16.0; Microsoft, Redmond, Washington, D.C., USA). During each trial, participants were instructed to maintain a power output of 20 W for a total of 40 s. This was considered to be a moderate paddle intensity based on previous studies (Schram, Hing & Climstein, 2016c).

Vicon data were visually inspected and labeled using Vicon Nexus 2.5 (Nexus; Oxford Metrics, Inc.. Yarnton, Oxfordshire, UK). Small gaps were filled using a built-in spline interpolation function with larger gaps filled using the pattern fill function (based upon the closest available anatomical landmark). Raw data files were exported from Nexus and analyzed in Visual3D (C-Motion, Germantown, MD, USA). Kinematic variables of interest included peak and minimum joint angles of the shoulder, elbow, lumbar spine and hip. Joint angle time series data were then analyzed using custom routines written in MATLAB (R2015b; Natick, MA, USA). The beginning and end of each stroke was defined as the maximum anterior position of the right hand on the right side and the left hand while paddling on the left side. These positions were taken from the raw marker data, and their respective times were then used to analyze joint angles. Range of motion (ROM), peak and minimum joint angles were then calculated from the mean stroke profile for each joint of interest. The shoulder, elbow, lumbar and hip mean joint angles were obtained by paddling on the left and right side and were then averaged together to generate a single profile of motion at the joint. This was achieved by combining the respective ipsilateral and contralateral angles (e.g., averaging the right shoulder during right side paddling with the left shoulder during left side paddling). The time series data were filtered (4th order Butterworth low pass, 20 Hz cut off) and averaged across each participants’ strokes. These average joint angle trajectories were plotted for comparison between levels of experience.

Data analysis

Descriptive statistics were calculated including means, standard deviation and coefficient of variance for each joint. Data were found to be not equally distributed in a Shapiro–Wilks test and therefore, Mann–Whitney U tests were conducted to determine differences between groups. Effect sizes were calculated for all comparisons to reflect the magnitude of change. The effect size was calculated using the method developed by Clark-Carter where the z value derived from the Mann–Whitney U Test is converted to into an r value (Clark-Carter, 2018). Consequently, the r value can be interpreted using the classification developed by Cohen, where an effect size of 0.1 could be considered small, 0.3 could be considered medium and 0.5 could be considered large (Cohen, 2013). Statistical significance was set at p = 0.05 and all statistical analyses were completed using the IBM Statistical Package for the Social Sciences (SPSS) v24.0 (SPSS Inc., Chicago, IL, USA).

Results

Experienced participants were found to be on average 9 years older (p < 0.001) than inexperienced participants. There were no other significant differences in height or weight between the groups. Table 1 shows the overall ROM, maximum and minimum joint angles for each for the assessed joints.

Table 1 Motion comparison at selected joints between inexperienced and experienced paddlers.

Results reported as average range ± SD.

Movement	Variable	Inexperienced	Experienced	U	Effect size	
Shoulder flexion	ROM	78.9 ± 24.9	56.6 ± 17.3	0.010*	−0.49	
Max	114.1 ± 23.5	131.5 ± 9.0	0.073	−0.36	
Min	35.2 ± 28.5	74.9 ± 16.3	0.001*	−0.62	
Elbow flexion	ROM	47.1 ± 22.0	47.7 ± 18.6	0.910	−0.28	
Max	68.5 ± 24.6	53.7 ± 21.7	0.152	−0.29	
Min	24.8 ± 13.5	6.0 ± 9.2	0.000*	−0.65	
Trunk flexion	ROM	5.4 ± 1.8	5.7 ± 1.6	0.572	−0.12	
Max	10.6 ± 8.3	12.18 ± 5.9	0.534	−0.13	
Min	5.2 ± 7.6	6.48 ± 6.5	0.497	−0.14	
Trunk abduction	ROM	9.3 ± 4.0	6.8 ± 1.8	0.055	−0.38	
Max	3.6 ± 3.6	1.7 ± 4.9	0.169	−0.28	
Min	−5.7 ± 3.8	−5.1 ± 5.8	1.000	−0.005	
Trunk rotation	ROM	43.4 ± 10.2	39.9 ± 9.8	0.427	−0.16	
Max	23.1 ± 10.0	19.1 ± 6.4	0.209	−0.25	
Min	−20.4 ± 9.8	−20.8 ± 7.3	0.910	−0.02	
Hip flexion	ROM	50.0 ± 18.5	66.4 ± 11.8	0.035*	−0.41	
Max	130.5 ± 14.9	134.2 ± 8.9	0.692	−0.09	
Min	80.6 ± 22.0	67.8 ± 6.8	0.055	−0.38	

Inexperienced participants demonstrated a significantly (p = 0.010) greater overall ROM in the shoulder compared with the experienced participants (78.9° ± 24.9° vs 56.6° ± 17.3° respectively); resulting in a 39.4% difference. Consequently, the minimum shoulder angle was significantly (p = 0.001) lower within the inexperienced participants compared with the experienced participants (35.2° ± 28.5° vs 74.9° ± 16.3° respectively); resulting in a 53% difference.

During hip flexion, the experienced participants demonstrated a significantly (p = 0.035) greater total ROM compared with the inexperienced participants (66.4° ± 11.8° vs 50.0° ± 18.5° respectively); resulting in 24.7% difference.

During elbow flexion the minimum angle was significantly (p = 0.000) less within the experienced participants (6.0° ± 9.2° vs 24.8° ± 13.5° respectively); resulting in a 75.8% difference.

Figure 1 displays these differences graphically. The experienced paddler is seen to display more hip flexion during the three stroke phases and less elbow flexion.

Figure 1 A graphical representation of the differences in stroke kinematics between experienced and inexperienced participants.

The experienced paddler at Entry (A), Drive (B) and Exit (C), and the inexperienced paddle at entry (D), Drive (E) and Exit (F).

Discussion

To our knowledge, this is the first known study to examine the stroke kinematics of SUP. The purpose of this research was to compare the differences in stroke kinematics between experienced and inexperienced participants. The results conclude that important differences exist in the paddling technique of both experienced and inexperienced participants, specifically at the shoulder, elbow and hip.

Previous research examining different skill levels in dragon boat racing found no differences in stroke kinematics between elite and sub-elite participants (Ho, Smith & O’Meara, 2009). In that study both the elbow and shoulder were examined during the entry, drive and exit phases of the stroke. It should be noted however, the reference group in the study were sub-elite experienced participants and not the inexperienced participants utilized in the current study. Paddling kinematics in the study highlighted within the reference group, 103° of elbow ROM and approximately 140° of shoulder ROM throughout the stroke cycle (Ho, Smith & O’Meara, 2009). Kinematic investigations of the kayak stroke have also reported elbow ROM in the order of 100° during the paddle stroke cycle amongst a variety of skill levels (Limonta et al., 2010). The fact that SUP is performed in a standing position would negate the need for larger shoulder and elbow ROM, highlighted by increased trunk flexion in the experienced group. This is thought to be a strategy to increase stroke length among experienced participants, who have previously been reported to have a longer, more powerful stroke than their more novice counterparts (Schram, Hing & Climstein, 2016c).

Overall, inexperienced participants displayed greater overall total shoulder ROM and less total hip flexion ROM while paddling. The reduced hip motion, combined with greater shoulder movement, illustrates a tendency for the inexperienced group to rely heavily on the shoulder and possibly the biceps and forearm musculature to generate force during the entry and drive phases of the stroke. It is unclear at this stage whether this may predispose the shoulder and elbow to injury, both of which were shown to be a common site of injury in SUP paddlers (Furness et al., 2017). In contrast, the experienced participants had less overall shoulder ROM and greater hip ROM. Interestingly, experienced participants initiated and ended the entry phase at a greater shoulder flexion angle, likely reflective of the greater hip flexion and a more horizontal trunk at the point of entry. Further, data indicated significantly less minimum elbow flexion in the experienced group, indicating the experienced participants were more likely to enter and drive through the stroke with an extended arm.

In summary, these data suggest that experienced participants rely less on shoulder and minimum elbow ROM but employ more hip flexion ROM. This would suggest a strategy facilitating an increased reaching motion before the initial paddle entry—a finding which may be of significance when considering the shoulder and elbow joints as injury prone regions in SUP participants (Furness et al., 2017).

Study limitations

This study was performed on an ergometer designed to simulate SUP paddling in the laboratory. While this ergometer has been shown to be a respectable surrogate for paddling in water (Schram, Hing & Climstein, 2016a), there are differences nevertheless. In particular, the ergometer does not account for water or wind conditions, which apply external perturbations to the board and can result in instability for the paddler. Therefore, postural control and balance related challenges were likely not adequately simulated with the ergometer. In addition, the cable and pulley system include a recoil mechanism that may provide a small amount of assistance during the recovery phase of the stroke. While this study may be an initial step at characterizing the kinematics of the SUP stroke, the results should be viewed with caution considering these differences.

The sample size and heterogenicity of the participants may have also affected the outcome of this analysis. Although 26 participants were included, only seven were experienced participants. Future studies should focus on a larger sample of experienced participants. Additionally, the current study failed to account for differences in handedness. Previous kayaking research has found differences in strength and co-ordination between dominant and non-dominant sides of the body (Kendal & Sanders, 1992) and future studies should also consider this variable. Finally, the inexperienced participants analyzed represented a wide range of experience levels, ranging from minimal exposure to SUP to 6 months experience at a recreational level. This led to a largely heterogenous group for the inexperienced participants. Some of the inexperienced participants also had difficulty maintaining the required power output for the duration of the assessments, consequently, differences in workload among the participants may have also affected the analyzed kinematics.

Conclusions

The results of this study suggest there are significant differences in paddle stroke kinematics between experienced and inexperienced SUP participants. Inexperienced participants appear to be more reliant on larger ranges of motion at the shoulder joint and less hip motion. Experienced participants appear to utilize less total shoulder ROM and more overall hip ROM. Identifying these different kinematic strategies may be of benefit for coaches, rehabilitation professionals and participants interested in improving technique and minimizing injury risk.

Supplemental Information

Supplemental Information 1 A graphical representation of the differences in stroke kinematics between experienced and inexperienced participants.

Click here for additional data file.

Supplemental Information 2 Ergometer set up in the laboratory and marker placement for Vicon plug in gait.

Click here for additional data file.

Supplemental Information 3 Stroke Comparison.

Click here for additional data file.

Supplemental Information 4 De-indentified raw data.

Click here for additional data file.

Additional Information and Declarations

Competing Interests

Author Contributions

Human Ethics

Ethics

Data Availability

The authors declare that they have no competing interests.

Ben Schram conceived and designed the experiments, performed the experiments, analyzed the data, contributed reagents/materials/analysis tools, prepared figures and/or tables, authored or reviewed drafts of the paper, approved the final draft.

James Furness conceived and designed the experiments, performed the experiments, analyzed the data, contributed reagents/materials/analysis tools, prepared figures and/or tables, authored or reviewed drafts of the paper, approved the final draft.

Kevin Kemp-Smith analyzed the data, prepared figures and/or tables, authored or reviewed drafts of the paper, approved the final draft.

Jason Sharp analyzed the data, prepared figures and/or tables, authored or reviewed drafts of the paper, approved the final draft.

Matthew Cristini analyzed the data, prepared figures and/or tables, authored or reviewed drafts of the paper, approved the final draft.

Daniel Harvie analyzed the data, prepared figures and/or tables, authored or reviewed drafts of the paper, approved the final draft.

Emma Keady performed the experiments, analyzed the data, prepared figures and/or tables, authored or reviewed drafts of the paper, approved the final draft.

Maichel Ghobrial performed the experiments, analyzed the data, prepared figures and/or tables, authored or reviewed drafts of the paper, approved the final draft.

Joshoa Tussler performed the experiments, analyzed the data, prepared figures and/or tables, authored or reviewed drafts of the paper, approved the final draft.

Wayne Hing conceived and designed the experiments, contributed reagents/materials/analysis tools, authored or reviewed drafts of the paper, approved the final draft.

Jeff Nessler conceived and designed the experiments, performed the experiments, analyzed the data, contributed reagents/materials/analysis tools, prepared figures and/or tables, authored or reviewed drafts of the paper, approved the final draft.

Matthew Becker analyzed the data, contributed reagents/materials/analysis tools, prepared figures and/or tables, approved the final draft.

The following information was supplied relating to ethical approvals (i.e., approving body and any reference numbers):

Permission to conduct the study was granted by the Bond University Human Research Ethics Committee (0000015422).

The following information was supplied relating to ethical approvals (i.e., approving body and any reference numbers):

The Bond University Human Research Ethics Committee granted Ethical approval to undertake this study within its facilities (0000015422).

The following information was supplied regarding data availability:

The raw data files are available in the Supplementary file; this data set show output from the kinematic testing undertaken in the biomechanics laboratory.

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
