# Peer review of "A biomechanical analysis of the stand-up paddle board stroke: a comparative study"

_PeerJ, doi:10.7717/peerj.8006_

## Round 0.1 · original submission · Minor Revisions

Dear authors
Sorry the delay in getting your paper revised. As you may know, we all work as volunteers for the journals and as an editor I have not much to do otherwise invite reviewers and wait. The specificity of the paper considering the PeerJ scope was also a difficult to find the reviewers. But now we have 3 good reviews on your paper and I invite you to carefully address the points mentioned and send a revised version together with a detailed response to the reviewer. Please highlight all changes made in the text.

Reviewer 1 ·

Basic reporting

The article is clear, very well written and relevant from the scientific point of view. It describes and compares the SUP technique, through articular angles, between experienced and inexperienced practitioners.
The references are adequate, but as it's a new issue to be investigated, there are really few references to be used. Table 1 is correct, but Figure 1 could be improved regarding its relation to the angular data found.

Experimental design

The experimental design is correct and it fits to the journal scope. Objectives are well defined. Methods were very well described and the study follows a high technical and ethical standard.

Validity of the findings

Data found are of relative importance, more related to the novelty of the study performed with SUP. Findings are robust, but effects size calculations are necessary to improve the discussion.

Additional comments

Congratulations on the work done and presented. However, in order to increase the quality of the study and better understand the results, I suggest some modifications:
In the abstract:
linha 27, você não pode "usar" seres humanos, eles eram "participantes".
linha 34: use a diferença "estatística" em vez de "significante". Eu sugiro que você faça essa mudança em todo o papel.
Introduction
line 48: are you sure that SUP is "easy to learn"? Maybe it's a very strong assertive.
Data analysis
line 146. Effects size of the groups is required, please see: J Exp Psychol Gen. 2012 Feb;141(1):2-18. doi: 10.1037/a0024338. Effect size estimates: current use, calculations, and interpretation. These results shoul be in Table 1.
Discussion
Line 177: "This is the first known study to examine the stroke kinematics of SUP."...Please be carefull, it's just to your knowledge.
Figure 1Although the Figure is very didactic, its design seems to indicate that the experienced ones have greater trunk flexion, which was not found.

Reviewer 2 ·

Basic reporting

The article is well written in a good scientific english.
The references are appropriated
I suggest to add a figure with the angular diagram of each analyses angle.
Please add the hypotheses.

Experimental design

The article has a clear aim.
The research question is simple and descriptive, but it is useful for coaches and others.
All the experiments were performed to a good technical standard.
The methods section is well written, but some correction are necessary.

Validity of the findings

This paper is a descriptive study.
The statistics is good
Some sentence in the discussion and conclusion should be reformulated to avoid possible misunderstandings.

Additional comments

COMMENTS TO THE AUTHORS

Dear authors, first of all, I would like to apologize for the time delay. I had some healthy problem during the last weeks and I did not have the possibility to work.

This manuscript describes a study undertaken to determine the kinematic differences between experienced and inexperienced stand-up paddle boarding participants. The authors use a 3D kinematical analysis to investigate the paddle technique on a specific ergometer. Results show that, inexperienced participants have larger shoulder ROM and a reduction in hip ROM compared with the experienced participants.
The manuscript is written well, but as you will see below, I have some methodological concerns.

MAJOR CONCERNS:

INTRODUCTION
The introduction is well written, but it is necessary to rearranged in order to improve the legibility.
In my opinion a simple structure could be useful:
- What is stan-up paddle boarding?
- Which are the positive benefits?
- Describe the biomechanics of SUP and the positive or negative alteration.
- Aims and hypothesis

As an example, in the first paragraph you defined the SUP and introduced the concept of SUP as positive “tool” to increase the heath of the participants.
However, at the beginning of the second paragraph, there is another description of the SUP.

Hence, in my opinion, could be useful to move the second paragraph on the first one.

Moreover, in the third paragraph (from line 75 to 89) you introduced your aim in the line 82, and reformulate the purposes in the line 86. Please, provide a clear statement of your aims at the end on the introduction.
Furthermore, please provide also your hypothesis.

METHODS
The methods section is well written and clear. No major concerns will be provide, but I have several minor comments (see below).

RESULTS
The results are well written and clear; however, I suggest to create an angular diagram to improve the clarity of the manuscript (see Nardello et al. 2019; Sports Biomechanics).

DISCUSSION
The discussion in well written, as the previously section. However, I suggest to highlight your speculation or consideration, as such.
As an example, Line 196: “The reduced hip motion, combined…”. The higher force generated by the biceps is a speculation, because you did not have the EMG or other kind of force measurements. Hence, please, reformulated all these kind of sentences.



SPECIFIC COMMENTS
INTRODUCTION:
Line 50: please add some healthy benefit of the SUP.
Line 82-84: “The purpose of these…”. You don’t have the possibility to state this, because you don’t have an injury or performance variable (e.g. velocity, power, mechanical stress…).

METHODS
Line 113: Provide some addition details regard the full Plug-In gait model, such as: the total number of markers and the number of body segments.

Line 116: Please provide the calibration parameters, such as the residuals. Are you sure that 100 Hz are sufficient to perform a good kinematic analysis? Do you have some references?

Line 127: “This was considered to be a…”. Did you have a measure of internal work, such as RPE or other kind of variables? I think that 20W are sufficient for the inexperience subjects and too easy for the experienced one.

Line 135: If you used the Plug-in gate, there is the possibility to obtain the joint angle from the Nexus software. Why did you choose to use a Matlab program?

Line 135: “The beginning and the end of each stroke…”. Did you use a threshold to identify this specific time instant or you visually defined this point?

Line 142: “The time series data were filtered…”. I suppose that you used a low-pass Butterworth filter, right? Please state which kind of filter did you use.
Moreover, if you use a 4th order Butterworth filter with a cut-off frequency of 20 Hz, how did you chose this frequency and order? I think that this filter is to strength, normally, a 2th order is sufficient.

FIGURE
Please provide an angular diagram of the analysed angle.

Reviewer 3 ·

Basic reporting

I´m not native speaker, but seems that the language is clear, intelligible and professional.
The article/references shows sufficient introduction and background.
The structure of the article shows acceptable format of ‘standard sections’
Raw data have been made available in accordance with data sharing policy.
Figures and tables
Fig 1 needs to be improved, better described and labelled, better resolution, also illustrating the range of motion (ROM) and peak joint angles.
Table 1 needs to be improved, showing also effect size and 95% CI.

Experimental design

The research was conducted rigorously and to a high technical standard and in conformity with the prevailing ethical standards in the field. It is an original primary research within aims and scope of the journal. Research question well defined, relevant & meaningful. The aim of this study was to compare the differences in SUP paddle stroke kinematics between experienced and inexperienced participants. The authors highlight that “findings may assist in identifying optimal stroke mechanics in order to minimize injury occurrence and improve overall performance.” However, confronting the inexperienced vs experienced rowing technique does not give us any assurance that performing the experienced technique will lower the incidence of injuries. What are the main characteristics in the paddling technique of injured subjects, whether in the shoulder, elbow or hip?
Line 127 (“This was considered to be a moderate paddle intensity based on previous studies (Schram et al. 2016c”): Heavy, severe and extreme intensities were not evaluated, why? Not relevant? Not especific? Is technique not different?
Since methods should be described with sufficient information to be reproducible by another investigator:
- Sample: could you show arm span, number of training sessions per week? Any dryland training sessions per week? Any physiological information (e.g. VO2)? Also, please clarify how many experienced and inexperienced male and female subjects; number of training sessions and dryland training per week…
- Please, show effect size as well as 95% CI for all comparisons;
- I strongly recommend one or two figures ilustrating the set-up (cameras and KayakPro SUPErgo) and study design. I´m sure you can use power point to make very nice figures;

Validity of the findings

'no comment'

Additional comments

The authors deserve to be congratulated for the effort of this work, which contributes to reduce the distance in science and practice. Works like this are very welcome, both in health and sports science. However, work is still needed to achieve the quality required for publication. Raw data can be opened, and is well described. I´m not native speaker, but seems that the language is clear, intelligible and professional.

Few comments
Line 42 (“These variations in strategy were…”): change “strategy” by “technique”.
Line 75 (“Currently, only anecdotal information”): Avoid jargon.

---

## Round 0.2 · accepted · Accept

I am writing to inform you that your manuscript - A biomechanical analysis of the stand-up paddle board stroke: A comparative study - has been Accepted for publication. Congratulations!